# COVID-19 and Cutaneous Squamous Cell Carcinoma—Impact of the Pandemic on Unequal Access to Healthcare

**DOI:** 10.3390/healthcare11141994

**Published:** 2023-07-10

**Authors:** Marko Jović, Milana Marinković, Branko Suđecki, Milana Jurišić, Zoran Bukumirić, Milan Jovanović, Milan Stojičić, Jelena Jeremić

**Affiliations:** 1Clinic for Burns, Plastic and Reconstructive Surgery, University Clinical Center of Serbia, 11000 Belgrade, Serbia; 2Faculty of Medicine, University of Belgrade, 11000 Belgrade, Serbia; 3Institute for Medical Statistics and Informatics, 11000 Belgrade, Serbia

**Keywords:** cutaneous squamous cell carcinoma, skin tumors, COVID-19, healthcare, accessibility to healthcare, public health, healthcare systems, lockdown

## Abstract

Most skin tumors are not fatal, but if not treated in a timely manner, they can lead to significant morbidity. Due to the COVID-19 pandemic and in order to create more capacities for the treatment of COVID-19-positive patients as well as to contain the spread of the virus, the healthcare system was reorganized worldwide, leading to decreased access to preventive screening programs. The aim of this study was to evaluate the impact of the pandemic on healthcare accessibility to cutaneous squamous cell carcinoma patients in Serbia. This retrospective study was conducted at the Clinic for Burns, Plastic, and Reconstructive Surgery, University Clinical Center of Serbia in Belgrade. Patient demographics and pathohistological findings of tumors of patients living in and outside the capital in the period before, during, and after the pandemic were compared. The two groups did not show any differences regarding the largest tumor diameter prior and during the pandemic; however, this difference became extremely noticeable after the pandemic (15 mm vs. 27 mm; *p* < 0.001). While cSCCs are commonly slow-growing tumors, the impact of the COVID-19 pandemic is not negligible. This study found a population at a significant risk of cSCC metastasis, with additional evidence likely to emerge in the upcoming years.

## 1. Introduction

Squamous cell carcinoma (cSCC) and basal cell carcinoma (BCC), also known as non-melanocytic skin carcinoma (NMSC), represent 95% of all skin tumors and are still one of the most common malignant tumors in the human population [1,2,3]. During 2019, about 2.4 million cSCC and 4.0 million BCC were reported worldwide [4]. According to the data from the Public Health Registry of the Republic of Serbia, 3545 cases of NMSC were registered, 1830 of which were in women and 1715 were in men [5]. While BCC is characterized by slow local tissue invasion and rare metastases, cSCC is found to be typically more aggressive in terms of both invasion and metastasis [6]. Treatment of patients with cSCC involves surgical excision with histopathological verification and further follow-up for at least 5 years after surgery. Despite patients with cSCC having a good prognosis after radical surgical excision, about 3.7% to 5.2% of patients are found to have nodal metastases, while 1.5% to 2.8% of patients have a fatal outcome [7,8,9]. Risk factors for poor outcome are a tumor diameter of more than 2 cm, poorly differentiated tumors, as well as perineural or perivascular invasion [3,7,8].

According to the World Health Organization (WHO) data, the first cluster of patients suffering from pneumonia was reported at the end of December 2019 by the Wuhan Municipal Health Commission from China, and coronavirus was mentioned as a potential cause for the first time [10]. On 13 January 2020, the first case of COVID-19 outside of China was reported in Thailand, while the pandemic was declared on 11 March 2020 [10]. According to official data from the World Health Organization (WHO), since the beginning of the COVID-19 pandemic, 762,201,169 COVID-19 cases have been confirmed in the world, with 6,893,190 deaths reported [11]. In the Republic of Serbia, 2,523,925 cases of COVID-19 have been registered, 17,996 of which had a fatal outcome [12].

Various public health principles and policies have since been implemented worldwide in efforts to contain the transmission of the virus. The first case of COVID-19 in the Republic of Serbia was reported on 6 March 2020, followed by the Declaration of a state of emergency by the National Assembly of the Republic of Serbia from 15 March 2020 to 6 May 2020 [12,13]. Due to the rapid increase in the number of COVID-19 patients worldwide, healthcare systems were reorganized in attempts to expand the capacities for treating COVID-19 patients. The reorganization involved transforming various healthcare institutions into COVID-19 hospitals, as well as recruiting all doctors, regardless of their specialty. Similar to emerging trends in the world, healthcare institutions in the Republic of Serbia have undergone major changes in order to respond to the challenges brought on by the pandemic.

The COVID-19 pandemic has significantly affected hospital admissions for non- COVID-19 patients around the world. While only urgent procedures were performed, all elective and routine diagnostic procedures, as well as routine follow-ups, were postponed and regulated by newly issued protocols [7,14,15,16,17,18]. O’Conell et al. reported a 42,8% reduction in patients admitted for general surgery between 1 March and 30 April 2020 [19,20]. Evaluating the effects of these postponements in oncology, colleagues from England found that even minor delays could lead to an over 30% reduction in survival at six months in incidental cancers as well as stage 2 or 3 tumors, i.e., the bladder, lung, esophagus, ovary, liver, pancreas, and stomach [21]. A similar situation was observed in internal medicine departments. In cardiovascular clinics throughout the USA and Europe, a decrease in the number of hospitalizations due to acute coronary syndrome as well as heart failure was recorded during the COVID-19 pandemic in comparison with the pre-pandemic period. For example, Cedrone et al. reported a reduction in hospital admissions for all cardiovascular diseases (CVDs) in Abruzzo, Southern Italy [22,23,24,25].

The aim of this study was to evaluate the impact of the COVID-19 pandemic on healthcare accessibility to cutaneous squamous cell carcinoma patients during the pandemic through an investigation into demographic data and tumor histopathology, as well as to determine the effects of these intermittent changes in hospital admission and surgery on cSCC patients in Serbia.

## 2. Materials and Methods

### 2.1. Study Design and Data Source

A retrospective cohort study was conducted at the Clinic for Burns, Plastic, and Reconstructive Surgery, University Clinical Center of Serbia, in Belgrade. The study included patients surgically treated at the Institution between 15 March 2019 and 31 December 2022. All included patients were treated for cutaneous squamous cell carcinoma according to the current AJCC guidelines, while exclusion criteria for this study were patients with individualized treatment protocols due to more specific characteristics, such as advanced disease prior to the pandemic.

After the approval of the Institutional Review Board (approval number 503/22), all data were extracted from patient medical records and histopathology reports and analyzed by the Pathology Department of the Medical Faculty, University of Belgrade. The data included patient demographics (age, sex, place of residence), lesion localization on the body, number of lesions per patient, as well as relevant histological findings (largest lesion diameter, disease form, cSCC histology grade, depth of invasion both in mm and by skin layer).

### 2.2. Period Definitions

Enrolled samples were divided according to the COVID-19 pandemic time frame into three categories: (1) pre-pandemic (samples obtained 15 March 2019–14 March 2020), (2) pandemic (samples obtained 15 March 2020–31 March 2022), and (3) post-pandemic (samples obtained 1 April 2022–31 December 2022). The pandemic time frame was set from the official beginning date of the lockdown in Serbia, 15 March 2020, until the recruitment of healthcare workers was over, and the clinic resumed its pre-pandemic work dynamic at the end of March 2022. The pre-pandemic and the post-pandemic periods were defined as the periods before and after the designated pandemic period.

### 2.3. Data Collection

Based on their reported place of residence, all patients were divided into two groups: (1) Capital and (2) non-Capital. The Capital group included all patients who territorially belong to the capital. The non-Capital group consisted of all patients who territorially belong to cities and villages outside the capital, and whose designated health institutions did not provide plastic surgery department.

Regarding the primary lesion localization on the body, all samples were divided into two major categories: (1) head and neck region; (2) other parts of the body. The reason behind this type of classification was to emphasize the sun-exposed areas vs. the non-exposed areas. All lesions were classified according to the disease form as: (1) Morbus Bowen or (2) invasive. The histologic grade of cSCC was categorized using the AJCC 2017 current guidelines (G1-4). The categorical depth layer of tumor invasion was classified using the histopathology report descriptions as: (1) in situ, (2) papillary dermis, (3) reticular dermis, (4) subcutaneous tissue, and (5) muscle.

### 2.4. Data Analysis

Results were presented as frequency (percent), median (range), and mean ± sd. For parametric data independent samples, a *t*-test was used to test the differences between groups. For numeric data with non-normal distribution and ordinal data, the Mann–Whitney U test was used. A Chi-square test or Fisher’s exact test was used to test differences between nominal data (frequences). All *p* values less than 0.05 were considered significant. Statistical data analysis was performed using IBM SPSS Statistics 22 (IBM Corporation, Armonk, NY, USA).

## 3. Results

### 3.1. Patient Demographics

In this study, 597 patients with 701 lesions treated at our facility were enrolled. The total characteristics of the patients included in the study are presented in Table 1. Of all the obtained samples, 219 were collected prior, 306 during, and 176 after the pandemic. The majority of them originated from the Capital group (n = 575, 82.0%) during all three periods (*p* = 0.732), while the rest comprised the non-Capital group (n = 126, 18.0%). The mean age of enrolled patients throughout all study periods was 75.91 ± 10.2. The non-Capital group had younger patients in the post-pandemic period compared to the Capital group (70.55 vs. 76.66; *p* = 0.005). During the first two time periods, women in the non-Capital group sought medical attention far more often than men compared to the Capital group (62.8% vs. 39.8%, *p* = 0.006; 51.9% vs. 32.3%, *p* = 0.007, respectively). This trend did not continue throughout the post-pandemic period (*p* = 0.935). No other significant differences regarding general demographic data between the two groups were observed.

### 3.2. Lesion Localization and Histological Findings

The vast majority of included patients (96.5%) reported this as their primary procedure on that lesion, with 74% of all treated patients having a single lesion. The non-Capital group had a notably higher percentage of multiple lesions per patient during the pre-pandemic period (25.8% vs. 9.3%; *p* = 0.016). However, this distribution difference was not observed throughout the other time periods (*p* = 0.485, *p* = 0.316, respectively) (Table 1).

Prior to the pandemic, lesions localized in the head and neck region occurred more often in the non-Capital group (88.4% vs. 68.8%, *p* = 0.010), while during the other two time periods, this difference was not observed (*p* = 0.399, *p* = 0.184, respectively). A detailed depiction of demographic data and histology findings in all three time periods is represented in Table 1.

When it comes to the tumor size, the two groups did not show any differences regarding the largest tumor diameter prior to the pandemic (*p* = 0.876). However, during the pandemic, this difference neared statistical significance (13.5 mm vs. 15 mm; *p* = 0.057) and ultimately became extremely noticeable after the pandemic (15 mm vs. 27 mm; *p* < 0.001) (Figure 1). Tumor depth did not correlate with the abovementioned changes in tumor diameter, with the two groups showing no significant differences in tumor thickness during and after the pandemic (*p* = 0.362; *p* = 0.314, respectively). However, the non-Capital group had notably thicker primary lesions prior to the pandemic (6 vs. 3 mm, *p* = 0.002).

In general, 71.7% of patients had the invasive form of cSCC, with 39% of lesions reaching the reticular dermal layer. Although no significant differences regarding disease form distribution amongst the two groups prior and during the pandemic were observed (*p* = 0.084, *p* = 0.176, respectively), non-Capital patients treated after the pandemic had a higher percentage of invasive cSCC (90.3% vs. 70.3%, *p* = 0.022) (Figure 1).

The majority of samples obtained during all three study periods were grade 1 (45.2%, 46.9%, 47.5%, respectively), with no other significant differences between the groups observed.

## 4. Discussion

In the course of 2020, 2021, and the first quartal of 2022 (until 31 March 2022), the Clinic for Burns, Plastic, and Reconstructive Surgery, Belgrade, Serbia, a tertiary healthcare institution, adjusted its work according to the official national recommendations and in accordance with the current epidemiological status. During the official lockdown, from 15 March 2020 to 6 May 2020, only severe conditions and life-threatening patients were hospitalized. In the following period, the work of outpatient surgery was suspended on several occasions, while the criteria for hospitalization were strict and defined according to the National Guidelines, with the aim of reducing the risk of virus transmission among patients and staff. Due to the dramatic epidemiological situation in the period lasting from 23 November 2020 to 1 February 2021, the clinic was temporarily converted into a COVID-19 hospital, with all plastic surgery patients triaged to other institutions. All these changes have disrupted the scheduled treatments of patients. As the number of COVID-positive patients with a severe clinical picture increased, anesthesiologists, surgeons, residents, and other medical staff were recruited to work in the University’s COVID-19 Hospital until 31 March 2022, when the clinic returned to its regular regime of work. Due to the insufficient number of healthcare workers available to treat non-COVID-19 patients and increased workload at the Clinic for Burns, Plastic, and Reconstructive Surgery as well as strict admission regulations, patients scheduled for elective surgery were intermittently triaged to other institutions, which had an especially large impact on patients who resided outside the capital. The inability of patients to perform regular skin examinations at dermatologists’ and plastic surgeons’ offices, as well as postponing elective surgeries, including oncological surgeries, on the one hand, as well as the fear of contracting COVID-19 on the other, resulted in larger skin tumors, possibly leading to worse outcomes and growing disease-specific mortality rates [3,17,26,27,28].

cSCC is known to be correlated to the cumulative lifetime UV exposure and commonly affects the elderly population [29]. While regarding the incidence of cSCC, the median age of primary cSCC was reported to be 78 years (IQR, 71–84 years) in male patients and 80 years (IQR, 71–87 years) in female patients in the literature. Age differences have been noticed for the pre-pandemic and post-pandemic non-Capital cohort (70.55 vs. 76.66; *p* = 0.005) [7,30]. This observation could be related to the inability of more elderly patients with more severe chronic illnesses to safely and effectively reach designated skin cancer tertiary institutions during the pandemic given the restricted use of public transportation. A further issue regarding cSCC patients lies in the fact that cSCC most commonly occurs in the elderly population, a vulnerable, high-risk group for COVID-19 infection [31]. In the COVID-19 era, every visit to the doctor represented a significant risk for virus transmission, especially for patients in high-risk groups such as the elderly, patients with comorbidities, or patients on immunosuppressive therapy [7]. The mean age found throughout all periods of this study was 75.91 ± 10.2 years. According to the available data, people aged 65–74 years had a 28.6% to 43.5% risk of hospitalization due to COVID-19 and a mortality rate of 2.7% to 4.9%, while people aged 75–84 had a 31.1% to 70.3% risk of hospitalization and a mortality rate of 4.3% to 10.5% [7]. Thus, the treatment decisions relied on doctors’ assessments of the risks of postponing oncologic surgery on the one hand and the potential risks of COVID-19 infection and its consequences on the other [7]. Moreover, given the age of the cSCC patients, being a high-risk group for COVID-19 complications, and their high association with chronic illnesses, these patients were more likely to have avoided going to the doctor, considering their skin changes insignificant.

In comparison with the pre-pandemic Capital resident patients, the pre-pandemic non-Capital resident patients had a notably higher percentage of multiple cSCC lesions per patient (25.8% vs. 9.3%; *p* = 0.016), with a predominance of lesions in the head and neck area (88.4% vs. 68.8%, *p* = 0.01), as well as larger tumor thickness (6 vs. 3 mm, *p* = 0.002). Distribution of lesion localization between the groups is presented in Figure 1. The number of lesions on the head and neck area is of great importance in expressing the significance of sun exposure in non-Capital patients. A large multicentric case–control study from the pre-pandemic period found outdoor workers to have a significantly increased risk of actinic keratosis, BCC, as well as cSCC [32]. However, the differences between the number, localization, and thickness of tumors per patient have not been observed between Capital and non-Capital residents in the pandemic and post-pandemic periods. This could be attributed to the triaging of patients with multiple lesions during the pandemic to other, non-tertiary institutions, as per the recommendations [7]. Invasive forms of cSCC are more commonly found in the head and neck area, while increased tumor thickness as well as localization on the ear are high risk factors for the metastasis of cSCC, giving these patients a priority [33,34]. Conversely, a conservative treatment, such as follow-ups, may have been elected for patients with in situ cSCC or tumors smaller than 2 cm by primary and/or secondary institutions after evaluating for the risk of COVID-19 infection complications, as recommended [7]. The National Comprehensive Cancer Network (NCCN), the British Association of Dermatologists (BAD), the British Society for Dermatological Surgery (BSDS), and the American College of Mohs Surgery (ACMS) have proposed guidelines for the treatment of skin tumors during the COVID-19 pandemic [7]. According to the guidelines, the treatment of patients with tumors smaller than 2 cm, well-differentiated, or in situ cSCC should be delayed for 3 months given that the risk of COVID-19 transmission and its consequences in this group of patients was greater compared to the potential consequences of oncological surgery delay. Conversely, patients with rapidly growing tumors, tumors larger than 2 cm in diameter, as well as ulcerated tumors with perineural and perivascular invasion had to be prioritized due to a higher risk of oncological progression [7]. Bearing in mind that the post-pandemic cohort included patients treated during the 8 months of the clinic’s regular work regime, in the upcoming period, we can expect an increase in the number of patients with multiple and neglected tumors.

Even though some protocols compensating for the reorganization of the healthcare system during the pandemic were effective, such as prioritizing patients with multiple or thicker lesions and efficiently triaging them to secondary healthcare institutions, thus reducing the number of these patients in the post-pandemic period, the outcomes of the conversion of specialized tertiary healthcare centers into COVID-19 hospitals are not without consequences. The tumor diameter in the non-Capital post-pandemic group was found to be significantly larger compared to the Capital post-pandemic group (15 mm vs. 27 mm; *p* < 0.001). Moreover, a growing trend of the tumor diameter was observed when comparing the non-Capital pandemic and the Capital pandemic group, nearing statistical significance (13.5 mm vs. 15 mm; *p* = 0.057), as shown in Figure 1. In comparison, the median diameter of the tumor in the pre-pandemic vs. post-pandemic non-Capital group was 13 mm vs. 27 mm, respectively, while for the same periods in the Capital group, the median diameters of tumors were 14 mm vs. 15 mm. Many of the patients residing in cities and villages without designated plastic surgery were most likely inadequately triaged to secondary institutions, leading to an increase in tumor diameter in unattended patients. Additionally, patients from the post-pandemic non-Capital group had statistically higher invasive forms of cSCC in comparison with post-pandemic Capital group patients (90.3% vs. 70.3%, *p* = 0.022), presented in Figure 1, suggesting that the Capital group may have had easier and quicker access to tertiary healthcare. The non-Capital patients were more likely to be referred to other secondary institutions, which shifted the high healthcare necessity pressure to the sub-optimally prepared institutions for such a high volume of patients, leaving many of these patients lost in the healthcare system.

While some authors found no impact of the delay from diagnostic to surgery on the size of tumors post-pandemic in countries such as the Netherlands, our results are in accordance with the findings of authors from Italy. [3,28] Cozzi et al. found a mean increase in the post-pandemic cSCC tumor diameter of 10.3 mm (95% CI 3–17.6). Additionally, the same authors found an absolute and percent increase in cSCC diagnoses in the post-pandemic group of patients [3]. The same was observed by Valenti et al., comparing skin tumor patients from 2019 to 2020, with an observed increase in patients with advanced NMSC [35]. During the COVID-19 pandemic, both patients and doctors faced new challenges regarding the availability of healthcare and the prioritization of treatment. Due to the impossibility of detecting skin tumors in a timely manner, malignant alteration of the precursors, the appearance of neglected tumors in terms of a larger diameter and thickness, as well as the appearance of metastases could be expected as a consequence [7,31].

Even though emerging evidence in the literature is conflicting regarding the impact of the pandemic on cSCC patients, the results of our study as well as the results of our colleagues are not negligible, given how the size of the tumor directly correlates to the increased chances for an unfavorable outcome. Most skin tumors are not fatal, but if not treated in a timely manner, they can lead to significant morbidity, low aesthetic outcomes, and high treatment costs [36,37]. Most importantly, tumors larger than 2 cm are known to be one of the described risk factors for metastasis of cSCC in the literature, as well as for an increase in the disease-specific mortality rate [3,7,8]. A treatment delay of more than 18 months has been indicated to lead to tumor growth of more than 2 cm [38]. Regardless of the type of skin cancer, preventive strategies, including regular dermatological examinations and dermoscopy, are of great importance in the early detection of cancer [39]. Due to the COVID-19 pandemic and in order to create more capacities for the treatment of COVID-19-positive patients as well as to contain the spread of the virus, the healthcare system was reorganized worldwide as well as in Serbia, leading to decreased access to preventive screening programs such as skin examination and routine dermoscopy. Aragón-Caqueo et al. found that the number of consultative dermatology examinations in Chile during 2020 decreased by 52.1% compared to 2019 (from 250,649 in 2019 to 120,095 in 2020) [15]. Tejera-Vaquerizo et al. also found that the number of patients in Spain treated for cSCC decreased by 44% during 2020 compared to the number of patients treated during 2019 (770 cases in 2019 and 429 cases in 2020) [40]. Similar results were obtained from the Netherlands, the USA, and the United Kingdom [28,41,42,43]. Valenti et al. proposed a causal relationship between the emergence of advanced skin tumors and the unavailability of dermoscopy, as well as regular follow-ups of previously diagnosed patients [35].

As the literature shows, various reports on the outcomes of specifically designed protocols for skin cancer patients during the pandemic yielded conflicting results, with many studies showing successful outcomes in triaging and treating these patients throughout the pandemic, while several others have provided proof of poorer outcomes. Overall results may seem successful from the treating physicians’ point of view, but from a wider, general healthcare availability point of view, a “one-protocol-fits-all” principle might not be applicable. Factors such as low- and medium-income countries (LMICs), socio-cultural factors, socio-economic factors, healthcare provider workload, or availability of transport and infrastructure all play a well-established role in healthcare accessibility [44]. One systematic review defined common potential factors for healthcare underutilization and/or inaccessibility in LMICs as well as in high-income countries (HICs). While patient experience played a more important role in healthcare utilization in HICs, in LMICs, barriers to healthcare access seemed to be more numerous and extreme even before the pandemic [44]. Extreme conditions such as the COVID-19 pandemic most certainly exacerbated the existing obstacles of healthcare availability and quality in LMICs. During the pandemic, understaffing and heavier workloads of healthcare workers could impede the proper triage and redirection of patients from designated institutions newly converted into COVID-19 hospitals to other secondary institutions, potentially underqualified in managing such high volumes of patients. Moreover, the timeline of the pandemic and the speed of virus transmission in a population, such as in Italy, may also play a role in healthcare readiness for such catastrophes, irrespective of the country’s income or other factors known to influence the accessibility and quality of healthcare. Nevertheless, immense amounts of effort have been put into providing care to all patients during the pandemic by all treating physicians worldwide.

Limitations of this study include the unavailability of accurate registers to compute the exact delay from the first appointment to treatment, as well as the patterns of rescheduling cancelled elective patients. This could provide useful information in determining the patients with more rapid tumor growth as well as the exact impact of the delay on the tumor diameter post-pandemic. Another limitation is lack of follow-up information on metastatic disease. These outcomes are to be expected in the upcoming years, and a follow-up study is deemed necessary. Other limitations include the potential biases commonly associated with the retrospective nature of the study.

## 5. Conclusions

While cSCCs are commonly slow-growing tumors, the impact of extreme conditions such as the COVID-19 pandemic is not negligible. Our findings add to an increasing body of data regarding the impact of the COVID-19 pandemic on cSCC patients and its influence on healthcare in general. This is noteworthy for several reasons. Firstly, our study evaluated the wholesome impact of the pandemic by choosing a specific timeframe that included all changes the healthcare system went through in the years following the declaration of the pandemic. Secondly, our results found a significantly larger tumor diameter after the pandemic, and cSCCs with a tumor diameter of 2 cm or more are at increased risk of disease-related outcomes. Finally, taking into consideration each and every result of different institutions in assessing their outcomes and carefully analyzing their differences is of utmost importance for the better preparation of potential similar future scenarios.

## Figures and Tables

**Figure 1 healthcare-11-01994-f001:**
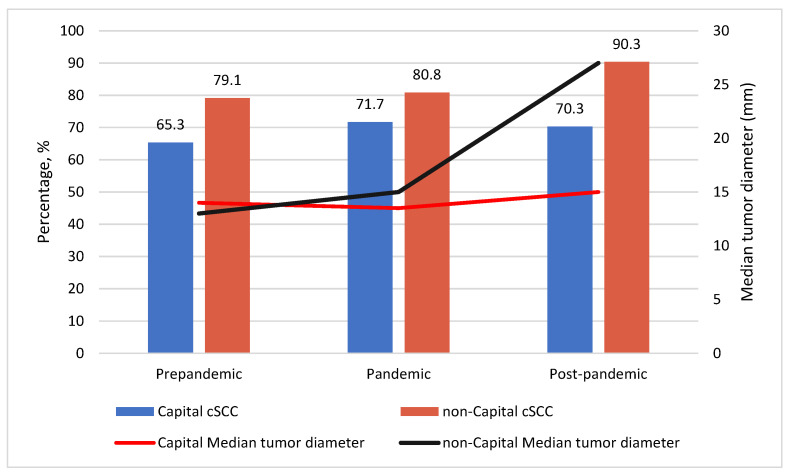
Distribution of invasive cSCC between the Capital and non-Capital group and tumor diameter median values during the three studied periods.

**Table 1 healthcare-11-01994-t001:** Patient demographic and histologic characteristics; n* = 597—number of included patients in the study; n** = 492—number of samples with available data; n*** = 682—number of samples with available data.

	Total	Pre-Pandemic n = 219			Pandemic n = 306			Post-Pandemic n = 176		
		Capital n = 176	Non-Capital n = 43	*p* Value	Capital n = 254	Non-Capital n = 52	*p* Value	Capital n = 145	Non-Capital n = 31	*p* Value
Age, mean ± SD	75.9 ± 10.2	76.2 ± 10.8	75.16 ± 8.8	0.533	76.0 ± 9.6	76.2 ± 8.7	0.933	76.6 ± 10.6	70.5 ± 12.3	0.005
Sex, n* (%)				0.006			0.007			0.935
Male	428 (61.1%)	106 (60.2%)	16 (37.2%)		172 (67.7%)	25 (48.1%)		90 (62.1%)	19 (61.3%)	
Female	273 (38.9%)	70 (39.8%)	27 (62.8%)		82 (32.3%)	27 (51.9%)		55 (37.9%)	12 (38.7%)	
First procedure on lesion, n (%)				0.586			1.000			0.660
Yes	677(96.6%)	173 (98.3%)	42 (97.7%)		245 (96.5%)	50 (96.2%)		138 (95.2%)	29 (93.5%)	
No	24(3.4%)	3 (1.7%)	1 (2.3%)		9 (3.5%)	2 (3.8%)		7 (4.8%)	2 (6.5%)	
Lesions per patient, n* (%)				0.016			0.485			0.316
One	519 (86.9%)	147 (90.7%)	23 (74.2%)		184 (85.2%)	41 (89.1%)		104 (88.9%)	20 (80.0%)	
More	78 (13.1%)	15 (9.3%)	3 (25.8%)		32 (14.8%)	5 (10.9%)		13 (11.1%)	5 (20.0%)	
Body localization, n (%)				0.010			0.399			0.184
Head and Neck	517 (73.7%)	121 (68.8%)	38 (88.4%)		186 (73.2%)	41 (78.8%)		105 (72.4%)	26 (83.9%)	
Other regions	184 (26.3%)	55 (31.3%)	5 (11.6%)		68 (26.8%)	11 (21.2%)		40 (27.6%)	5 (16.1%)	
Tumor thickness, median (range)	4.0 (2–41)	3 (0.2–25)	6 (0.5–20)	0.002	4 (0.5–40)	3.5 (0.5–25)	0.362	3 (0.5–41)	3 (0.5–24)	0.314
Disease form, n (%)				0.084			0.176			0.022
Morbus Bowen	198 (28.2%)	61 (34.7%)	9 (20.9%)		72 (28.3%)	10 (19.2%)		43 (29.7%)	3 (9.7%)	
Invasive	503 (71.8%)	115 (35.3%)	34 (79.1%)		182 (71.7%)	42 (80.8%)		102 (70.3%)	28 (90.3%)	
Histology grade, n** (%)				1.000			0.528			0.418
G1	229 (46.5%)	52 (45.2%)	14 (45.2%)		84 (46.2%)	21 (50.0%)		45 (46.9%)	13 (50.0%)	
G2	185 (37.6%)	38 (33.0%)	11 (35.3%)		68 (37.4%)	16 (38.1%)		39 (40.6%)	13 (50.0%)	
G3	48 (9.8%)	20 (17.4%)	3 (9.7%)		17 (9.3%)	3 (7.1%)		5 (5.2%)	0 (0.0%)	
G4	30 (6.1%)	5 (4.3%)	3 (9.7%)		13 (7.1%)	2 (4.8%)		7 (7.3%)	0 (0.0%)	
Invasion depth layer, n*** (%)				0.097			0.183			0.103
In situ	200 (29.3%)	62 (36.5%)	10 (23.8%)		71 (28.2%)	11 (21.6%)		43 (31.2%)	3 (10.3%)	
Papillary dermis	58 (8.5%)	16 (9.4%)	4 (9.5%)		24 (9.5%)	4 (7.8%)		6 (4.3%)	4 (13.8%)	
Reticular dermis	274 (40.2%)	59 (34.7%)	17 (40.5%)		101 (40.1%)	22 (43.1%)		62 (44.9%)	13 (44.8%)	
Subcutaneous tissue	86 (12.6%)	20 (11.8%)	5 (11.9%)		37 (14.7%)	6 (11.8%)		11 (8.0%)	7 (24.1%)	
Muscles	64 (9.4%)	13 (7.6%)	6 (14.3%)		19 (7.5%)	8 (15.7%)		16 (11.6%)	2 (6.9%)	

## Data Availability

Data can be accessed from corresponding author upon request.

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
