# Peer review of "COVID-19 and Cutaneous Squamous Cell Carcinoma—Impact of the Pandemic on Unequal Access to Healthcare"

_healthcare, 2023, doi:10.3390/healthcare11141994_

Round 1

Reviewer 1 Report

This is an exellent review and explanation of the state of SCC in Serbia a cancer we do not often think of. I commend the authors for this work

They wondered if COVID has an impact on the presentation of SCC and it did. The tumours all got larger and thus potentially more dangerous.  It was extremely well written and described their local pandemic situation extremely well and very eloquently.  

THis is mainly a descriptive paper,  There is not revolutionary finding,  It is up to you if you thing this has value for readers to read and see and if it will have an impact to get policy makers to manage a pandemic in a different fashion in the future to prevent such delays to access to care to improve cancer outcomes.  

It was a very well written and strightforward paper asking a simple question with a simply answer no complex stats required or anything this is why I accepted it.  

It was an interesting read. Not mega science but a true commentary on we are all now seeing post pandemic with delayed care and more advanced cancers. This is just the proof that it is not just anecdotal, the data supports it. 

Author Response

Dear Sir/Madame,

We want to thank You for Your time and comments. We are very grateful that you consider our work worth reading. 

Kind regards.

Assistant Professor

Marko Jović MD, PhD

University Clinical Center of Serbia

Reviewer 2 Report

The work of the authors is very interesting to my opinion because it deals with an important topic of public health, the delay in diagnosis and treatment of cancer because of Covid. Moreover, the authors report extensively on the experience of a small country with historical difficulties and multicultural reality, and this is very appreciated.

I perceive the introduction as too long and I suggest some shortening with more concise formulation of points. At the same time, if available, it would be of interest to add data on the economic background of patients, if it could have had an impact, and a more deep discussion of gender ans ethnic differences if possibile.

In the methods section, there are some typos, e.g. lines not aligned, the statistical part is somehow divided from the rest: I can suggest to report the methods in subparagraph mode

Only minor polishing of language and removal of redundancies.

Author Response

Dear Sir/Madame,

We want to thank You for Your time and comments. We are very grateful that you consider our work worth reading. According to the recommendations, we changed the following items in the manuscript:

  1. As recommended, we tried to shorten the introduction. We believe that the information presented in the introduction is very important for readers to better understand the need for our research. The only way to shorten the introduction without losing the quality of the paper was to displace the paragraph that begins with the sentence In the course of 2020, 2021 and the first quartal of 2022 (until March 31st, 2022) the Clinic for Burns, Plastic and Reconstructive Surgery, Belgrade, Serbia, a tertiary healthcare institution, adjusted its work according to the official national recommendations and in accordance with the current epidemiological status...“ to the beginning of the discussion. Given that this paragraph is a significant part of the manuscript that represents the organization of health care and the way the clinic works during the pandemic, we believe that it needs to be described in detail. If you feel it has no place in the discussion, we would be grateful if the mentioned paragraph remains in the introduction.
  2. Your suggestion to add data on the economic background of patients and ethnicity is great. Unfortunately, considering that the study was designed as a retrospective one, we do not have that data. This can be an excellent idea for future research. On the other hand, we presented our results about gender as You suggested: During the first two time periods, women in the non-Capital group sought medical attention far more often than men, when compared to the Capital group (62.8 % vs. 39.8 %, p=0.006; 51.9% vs. 32.3%, p=0.007, respectively). This trend did not continue throughout the post-pandemic period (p=0.935).
  3. We report the Materials and Methods section in subparagraph mode as You suggested.
  4. We did minor polishing of language and removal of redundancies as You indicated.

Kind regards.

Assistant Professor

Marko Jović MD, PhD

University Clinical Center of Serbia

Reviewer 3 Report

Thank you for the opportunity to review this article, which aims to analyse the impact of the COVID-19 pandemic on sSCC outcomes. Please note my considerations:

1-Please use references that support the information provided. Check this for all references. For example, “…,about 3.7% to 5.2% of patients are found to have nodal 41 metastases, while 1.5% to 2.8% of patients have a fatal outcome [7].”

Reference 7 only mentions other references from which the data were obtained. In fact, one of them is listed in the bibliography of the document.

Eigentler TK, Leiter U, Hafner HM, Garbe C, Rocken M, Breuninger H. Survival of patients with cutaneous squamous cell carcinoma: results of a prospective cohort study. J Invest Dermatol. 2017;137:2309‐2315. 

Thompson AK, Kelley BF, Prokop LJ, Murad MH, Baum CL. Risk factors for cutaneous squamous cell carcinoma recurrence, metastasis, and disease‐specific death: a systematic review and meta‐analysis. JAMA Dermatol. 2016;152:419‐428. 

2-The tables are repetitive. They do not provide enough information to constitute three tables.

Table 1 can be included in table 3, just by putting after the columns for capital and extra-capital the total of both. Table 2 can be included in table 3, just by adding a heading "multiple injury N (%)".

If Figure 1 is retained, In Table 3 can be deleted. “Body localization, Head & Neck, Other regions”

If Figure 2 is retained, In Table 3 can be deleted “Largest tumor diameter, median (range)”, de a tabla 3

The caption text for tables and figures should clearly reflect what is being shown but should not repeat reasoning about the data it shows that is already repeated in the main text. I advise eliminating these repetitions.

3-I'm afraid I don't understand this sentence very well, and the same may happen to eventual readers. Could the authors rephrase it more clearly?

“Nevertheless, 272 given how the post-pandemic cohort was comprised of patients in an 8-month span, an 273 increase in patients with cSCC in sun-exposed areas as well as multiple tumors could 274 possibly be expected in the upcoming period, and the effects of the newly adopted pro-275 tocols during the pandemic are yet to emerge.”

4-I strongly recommend not to overuse terms that have a specific scientific meaning and, if used generically, lead readers to confusion. For instance:

- “evidence”: “Additionally, patients from the post-pandemic non-Capital group 292 had statistically higher invasive forms of cSCC in comparison with post-pandemic Cap-293 ital group patients (90.3% vs 70.3%, p=0.022), presented in Figure 2, providing evidence 294 of easier and quicker access to tertiary healthcare by Capital residents.

Instead “evidence” it is more accurate to use, “…Figure 2, suggesting that Capital group may have had easier and quicker access to tertiary health care.”

-“Significant”:  “Secondly, our results found a population at a 375 significant risk of cSCC metastasis”. It will be more accurate "Secondly, our results found a significantly larger tumour diameter after pandemic, and cSCCs with tumour diameter of 2 cm or more are at increased risk of disease-related outcomes.”, Only if this is the message the authors want to convey

Author Response

Dear Sir/Madame,

We want to thank You for Your time and comments. We are very grateful that you consider our work worth reading. According to the recommendations, we changed the following items in the manuscript:

  1. We have checked all references. We have corrected it according to Your recommendations. We added references number 36. and 37.

- 36. Eigentler, T.K.; Leiter, U.; Häfner, H.-M.; Garbe, C.; Röcken, M.; Breuninger, H. Survival of Patients with Cutaneous Squamous Cell Carcinoma: Results of a Prospective Cohort Study. J. Invest. Dermatol. 2017, 137, 2309–2315, doi:10.1016/j.jid.2017.06.025.

- 37. CDC COVID19 Response Team . Severe outcomes among patients with coronavirus disease 2019 (COVID19)–United States, February 12March 16, 2020. MMWR Morb Mortal Wkly Rep. 2020, 69, 343346.

  1. As per Your suggestion, we incorporated the data from Tables 1 and 2 into Table 3 and removed Figure 1. The caption text for both supplements has also been revised. (Pages 4/13 and 5/13 in the manuscript.)
  2. We corrected items numbered 3 and 4 according to the recommendations. We rephrased the sentences to make them more understandable for readers:

- Instead of “Nevertheless, given how the post-pandemic cohort was comprised of patients in 8 months, an increase in patients with cSCC in sun-exposed areas as well as multiple tumors could possibly be expected in the upcoming period, and the effects of the newly adopted protocols during the pandemic are yet to emerge” we wrote “Bearing in mind that the post-pandemic cohort included patients treated during the 8 months of the clinic's regular work regime, in the upcoming period we can expect an increase in the number of patients with multiple and neglected tumors”

- Instead of “Additionally, patients from the post-pandemic non-Capital group had statistically higher invasive forms of cSCC in comparison with post-pandemic Capital group patients (90.3% vs 70.3%, p=0.022), presented in Figure 2, providing evidence of easier and quicker access to tertiary healthcare by Capital residents” we wrote “Additionally, patients from the post-pandemic non-Capital group had statistically higher invasive forms of cSCC in comparison with post-pandemic Capital group patients (90.3% vs 70.3%, p=0.022), presented in Figure 2, suggesting that Capital group may have had easier and quicker access to tertiary health care. ” 

- Instead of Secondly, our results found a population at a 375 significant risk of cSCC metastasis”, we wrote, "Secondly, our results found a significantly larger tumour diameter after the pandemic, and cSCCs with a tumor diameter of 2 cm or more are at increased risk of disease-related outcomes.” 

Kind regards.

Assistant Professor

Marko Jović MD, PhD

University Clinical Center of Serbia